# Impact of Glucocorticoid Use in Oncology in the Immunotherapy Era

**DOI:** 10.3390/cells11050770

**Published:** 2022-02-22

**Authors:** Laura Kalfeist, Loïck Galland, Fanny Ledys, François Ghiringhelli, Emeric Limagne, Sylvain Ladoire

**Affiliations:** 1Platform of Transfer in Cancer Biology, Georges-Francois Leclerc Center, 21000 Dijon, France; lkalfeist@cgfl.fr (L.K.); lgalland@cgfl.fr (L.G.); fledys@cgfl.fr (F.L.); fghiringhelli@cgfl.fr (F.G.); elimagne@cgfl.fr (E.L.); 2UMR INSERM 1231 “Lipides Nutrition Cancer”, 21000 Dijon, France; 3Department of Medical Oncology, Georges-François Leclerc Center, 21000 Dijon, France; 4School of Medicine, University of Burgundy Franche-Comté, 21000 Dijon, France

**Keywords:** cancer, corticosteroids, immunotherapy, immune checkpoint inhibitors, immune-related adverse event

## Abstract

Thanks to their anti-inflammatory, anti-oedema, and anti-allergy properties, glucocorticoids are among the most widely prescribed drugs in patients with cancer. The indications for glucocorticoid use are very wide and varied in the context of cancer and include the symptomatic management of cancer-related symptoms (compression, pain, oedema, altered general state) but also prevention or treatment of common side effects of anti-cancer therapies (nausea, allergies, etc.) or immune-related adverse events (irAE). In this review, we first give an overview of the different clinical situations where glucocorticoids are used in oncology. Next, we describe the current state of knowledge regarding the effects of these molecules on immune response, in particular anti-tumour response, and we summarize available data evaluating how these effects may interfere with the efficacy of immunotherapy using immune checkpoint inhibitors.

## 1. Background: The Historical Role of Glucocorticoids in Cancer Treatment

Thanks to their anti-inflammatory, anti-oedema, and anti-allergy properties, glucocorticoids (GCs) are among the most widely prescribed drug classes in patients with cancer. However, improvements in our understanding of their effects on immunity have raised the legitimate question of the impact of GCs use on cancer progression, especially since immunotherapy has emerged as a major treatment at the advanced stage in numerous forms of cancer.

In this review, we give an overview of the main current indications for GCs in patients with cancer. Next, we discuss current biological knowledge of the effects of GCs on the main components of the immune response (especially the components involved in anti-tumour immune response). Finally, in light of recently published clinical findings, we review the possible impact of GCs on the efficacy of immunotherapy with immune checkpoint inhibitors (ICI) in patients receiving these two types of treatment.

GCs currently occupy a central position in the therapeutic arsenal for the management of solid or haematological tumours (Figure 1). Schematically, GCs can be used for their direct anti-cancer effects, for their anti-inflammatory effects, or for prophylaxis (or treatment) of certain side effects of anti-cancer therapies. More recently, GCs have even been used to counter adverse, immune-mediated effects in patients treated by immunotherapy.

### 1.1. Effect of Glucocorticoids on Tumour Cells: Pro- or Anti-Tumour?

Evidence indicates that GCs induce apoptosis in haematological cells, thus supporting their use as chemotherapeutic agents for leukaemias, lymphomas, and myeloma. GCs are a therapeutic component in their own right in chemotherapy protocols for haemopathic malignancies and have been shown to be very efficacious in association with cytotoxic chemotherapy for the treatment of acute lymphoblastic leukaemia, chronic lymphocytic leukaemia, Hodgkin and non-Hodgkin lymphoma, as well as multiple myeloma [1]. The mechanisms by which GCs could induce tumour cell apoptosis are still incompletely understood but appear to be multiple and interrelated.

Indeed, tumour cell death appears to require GC-induced gene regulation. Regarding the potential cytotoxic effects, there have been reports that GCs are implicated in the activation of apoptosis in tumour cells, with activation of the genes capable of inducing pro-apoptotic molecules, such as Bim, and repression of the anti-apoptotic *BCL2* molecules [2]. Death receptor and mitochondrial apoptosis signalling as well as caspase activity could be enhanced for example by dexamethasone in lymphoid cells. Because of these pro-apoptotic properties in lymphoid tissues, GCs are frequently used as co-treatment [3]. Conversely, other signalling pathways involving AP-1 or NF-κB and key to cell proliferation processes are inhibited [4]. It is also accepted that GCs inhibit proliferation by suppressing *c-MYC* [4]. In parallel with this phenomenon, *GILZ*, a target transcriptional factor, is regulated by GCs and may induce apoptosis in multiple myeloma [5]. Benefits of GCs treatment could also come from the up-regulation of the thioredoxin-interacting protein (*TXNIP*), resulting in reactive oxygen species accumulation and apoptosis [6]. Lastly, concerning haematological malignancies, it has also been described that GCs can repress miR-17-9, an miRNA that correlates with apoptosis or affects the redox balance, especially in multiple myeloma cells, increasing the susceptibly to cell death.

Nevertheless, it is also important to note that mechanisms of resistance to GCs have been described in haematological malignancies, such as increased secretion of IL-6 in multiple myeloma [7], activating mutations of NOTCH [8], or overexpression of the Akt/mTor signalling pathway in acute leukaemia [9,10].

For the treatment of solid tumours, GCs are also used in association with cytotoxic chemotherapies [11] but mainly for prophylaxis of side effects, without any additional, direct anti-tumour effect having been clearly demonstrated. In in vitro models, inhibition of the process of invasion/migration of tumour cells has been described, involving different mechanisms, such as down-regulation of RhoA [12], MMP2/9, and IL-6, or via induction of E-cadherin [13,14]. Some authors have also underlined a beneficial effect on tumour neo-angiogenesis through the suppression of pro-angiogenic factors, such as VEGF or IL-8 [15]. More specifically in the context of breast or prostate cancer, there have been reports of an interaction between the GC receptor (GR) and oestrogen or androgen receptors that could limit tumour cell proliferation via a “hormone-therapy-like” effect [16,17,18]. GCs treatment was also shown to induce transcription of micro RNA, leading to a reduction of the metastatic process in murine models [19]. However, such biological effects of GCs in clinical conditions remain to be formally demonstrated. 

Indeed, in vitro or animal models have yielded conflicting results, with other teams suggesting that GCs induce a reduction in cell adhesion and stimulate cell motility, thereby potentially enhancing the metastatic risk [20,21]. GCs could also be associated with the activation of resistance mechanisms against chemo- or radiotherapy. Accordingly, an increase in IkB-alpha has been described, which inhibits the NF-kB pathway as well as an increase in SGK1 (serine/threonine survival kinase 1) [22,23]. However, these results must be taken with caution, especially because some studies in ovarian cancer did not appear to find any deleterious effect of GCs during chemotherapy on patient survival [24]. Thus, the implication of GCs in pro-tumour processes, notably tumour proliferation, cell adhesion, or epithelial–mesenchymal transition, remains the subject of controversy. It has also been suggested that the use of GCs may render tumours resistant or less susceptible to apoptosis after cancer therapy. Indeed, GC treatment was found to down-regulate basal and chemotherapy-induced expression of apoptosis effectors in human cervical and lung carcinoma cells [25]. Variations in effects have also been observed within tumours, and in breast cancer, variable effects were shown according to the histological subtype or according to the tumour microenvironment [21,26,27,28]. 

In an attempt to elucidate these discordant findings, summarized in Table 1, some authors have examined the clinical benefit of GCs on top of usual therapy in patients treated in routine practice [11,18]. The conclusions argue in favour of wide variability that depends on the type of cancer. GC treatment has been shown to be very effective in haematological malignancies There is also certain benefit to be yielded in breast or prostate cancer, which is, however, not replicated in digestive tract tumours, while there is even a potentially deleterious effect at high doses in lung cancers. 

### 1.2. Effect of Glucocorticoids on Cancer-Related Symptoms

In oncology, GCs have long been widely used to help manage cancer-related symptoms, especially those related to inflammation and/or oedema caused by the tumour. The majority of patients followed up in oncology will receive GCs at some point during the treatment of their cancer. One of the most commonly encountered symptoms is cancer-related fatigue, especially at advanced stages of the disease. The evidence in favour of the efficacy of corticosteroids for the management of cancer-related fatigue remains weak [29,30]. However, some authors report an improvement in the feeling of physical and mental well-being with the use of this therapeutic option.

Pain is another feared manifestation in cancer patients. Opioid therapy remains the treatment of choice, but corticosteroids are often used as co-analgesics. Thanks to their anti-inflammatory properties, corticosteroids have been reported to have some efficacy against cancer-related pain of inflammatory origin [31]. Corticosteroid therapy is recommended for the management of cancer-related pain, especially in bone metastasis [32]. However, some studies and meta-analyses have highlighted that the efficacy is less than expected and should be considered in perspective with the potential adverse effects of this treatment [29,33,34].

Dyspnoea is one symptom where corticosteroid therapy may be efficacious. Dyspnoea affects up to 70% of cancer patients at the end of life and is often multifactorial in origin. There are currently several arguments in favour of the use of corticosteroids, depending on the origin of the dyspnoea, in particular in conditions that arise during tumour progression, such as bronchospasm, pleural effusion, or superior vena cava syndrome [35,36,37]. GCs have anti-inflammatory activities, and this may explain their ability to attenuate dyspnoea in view of the elevated inflammatory response in patients. As described in chronic obstructive pulmonary diseases (COPD), cancers are characterized by a significant inflammatory component that includes immune airway wall infiltration, increased pro-inflammatory cytokines in lung tumours (such as IL-8, IL-6, C-reactive protein), or increased peripheral neutrophil activation [38,39]. However, as dyspnoea is likely to be multifactorial in the context of cancer, GCs may be more efficient in some cases, such as carcinomatous lymphangitis or airway obstruction by tumour, but less so in other circumstances.

Corticosteroids have become established as a key element in the management of symptomatic cerebral oedema [40] and are recommended to reduce intracranial pressure related to the progression of a primary brain tumour or intracerebral metastasis. Dexamethasone is recognized as the treatment of choice to achieve relief of symptoms of tumour progression, such as signs of intracranial hypertension (nausea, vomiting, headache) or neurological deficits [41,42]. Dexamethasone is often chosen because it also contributes to improving the Karnofsky index in these patients but also for its limited mineralocorticoid effects and the reduced risk of a rebound effect after discontinuation [43]. The management of leptomeningeal metastasis and its symptoms is generally more complex and may require intrathecal administration of GCs [44]. Similarly, corticoid therapy counts among the main efficacious therapeutic options in the management of symptoms related to spinal cord compression, notably during metastasis to the spinal cord occurring in numerous cancers, such as breast, prostate, lung, or kidney cancer [45,46].

GCs are often used with systemic administration to treat occlusive digestive symptoms of tumoral origin in the aim of restoring or maintaining bowel activity [47].

In addition to these specifically intended effects against cancer-related symptoms, GCs may also have favourable effects in advanced cancer patients on the overall state thanks to their orexigenic effects [29,48,49] as well as a positive impact on asthenia. Overall, several randomized trials have shown improved quality of life in patients receiving GCs in advanced cancer compared to placebo [29,50]. However, it should be noted that certain side effects of GCs in the clinical context (oral candidosis, insomnia, severe proximal myopathy) may hamper their use by outweighing any potential benefit [43,51]. Therefore, even in advanced cancer, the rule should be to use the lowest effective dose and for the shortest duration possible.

### 1.3. Effect of Corticosteroids on Treatment Side Effects

GCs are often prescribed to help manage the side effects of anti-tumour therapies, first among which are nausea and vomiting induced by chemo- or radiotherapy. Although GCs count among the longest-existing drugs used for the prevention of chemotherapy-related nausea and vomiting, they remain nonetheless firmly anchored in current recommendations in association with 5HT3 or NK1 receptor antagonists [52,53,54] according to the emetic response of the chemotherapy protocol. Indeed, chemotherapies induce the production of inflammatory mediators, such as eicosanoids, which may explain the effectiveness of GC treatments for preventing nausea and vomiting [55]. GCs may also reduce pain and opioid use, reducing opioid-related nausea [56]. Several other molecular mechanisms could be involved [57], such as their direct action on the solitary tract nucleus or regulation of the hypothalamic-pituitary-adrenal axis [58,59]. Interactions with certain neurotransmitters have also been suggested (serotonin, adrenaline, etc.) [60].

Certain types of chemotherapy (bleomycin, gemcitabine) [61] or targeted therapies, EGFR (erlotinib, gefitinib) [62], mTOR or MEK inhibitors [63], or even monoclonal antibodies targeting HER2 (especially certain new antibody-drug conjugates such as trastuzumab deruxtecan, known to cause interstitial lung disease (ILD)) can be responsible for lung toxicity [64]. In parallel to discontinuation of the implicated drug, many physicians also prescribe GCs, which can help to resolve the dyspnoeic symptoms and improve the pulmonary radiographic presentation [65]. Radiotherapy may also cause complications that can be alleviated by GCs (brain radionecrosis, radiation pneumonitis, etc.) [42,66].

Allergic reactions occurring during infusion of certain anti-cancer therapies constitute a medical therapeutic emergency. The incidence of hypersensitivity can be quite high for some agents, such as platinum salts or taxanes, but also for certain monoclonal antibodies. The main symptoms coincide with those of a hypersensitivity reaction or even anaphylactic shock and become apparent in the form of a rash or, in more severe cases, laryngeal oedema, dyspnoea, or hypotension. Mast cells and basophils play a central role in mediating an anaphylactic reaction. GCs are anti-allergic compounds that reduce the number, maturation, and activation of mast cells. They rapidly decrease histamine release and up-regulate anti-inflammatory mediators by a non-genomic mechanism, acting through membrane-bound or cytosolic receptors [67]. Pre-medication with GCs before administration of taxanes (docetaxel, paclitaxel) or of etoposide or asparaginase [68] makes it possible to reduce the risks of developing anaphylactic reactions [69,70,71]. In the case of a severe infusion reaction, GCs administered at high doses (1–2 mg/kg prednisolone equivalent) are the cornerstone of treatment along with anti-histamines [67,68].

Clearly, corticosteroids have become well established in the management of patients with cancer at several levels although primarily in the management of cancer-related complications or treatment side effects. Nonetheless, it should be underlined that the use of a number of therapeutic approaches using GCs [72] lack a solid scientific rationale even though the side effects of GCs treatment are well known. This is all the more important to keep in mind since the advent of immunotherapy, which is quickly becoming a leading component of therapy in the majority of cancers.

The effects of corticosteroid therapy on the immune system of patients with cancer deserve close attention but are actually not widely described. In particular, the possibility that corticosteroids might blunt the anti-tumour immune response and actually have a deleterious effect has been raised. The advent of immunotherapy using ICI and their rapid spread into the treatment of many tumour types has yielded marked and lasting response or even cure at the metastatic stage. Furthermore, their introduction at the early stages of disease, including as adjuvant therapy, highlights the need to adequately control the full spectrum of immunological effects of any other treatments being co-administered along with immunotherapy, notably GCs.

## 2. Effects of Glucocorticoids on the Immune System and the Immune Response in the Setting of Cancer

From a biological point of view, synthetic GCs, as for endogenous hormones, are liposoluble molecules that can cross the plasma membrane and bind their receptor within the cell. The GC receptor (GR, encoded by the *NR3C1* gene) is thus activated and translocates to the nucleus in the form of a homodimer, where it regulates gene transcription (stimulation or repression) by directly binding to glucocorticoid response elements (GREs) [73]. In the absence of a ligand, GR localizes in the cytoplasm in a multiprotein complex containing heat shock proteins, immunophilins, and other chaperones, which improve the receptor’s affinity for the ligand and prevent the degradation of the receptors.

On top of these direct genomic effects, the complex ligand-GR also has indirect genomic effects through direct interactions with other proteins (including some transcription factors, such as NF-κB [74,75] or AP1 [76,77], resulting in their inhibition). In this way, GR regulates their transcription via protein-protein interactions (Figure 2).

Overall, the biological effects mediated by the action of the GR comprise an increase in the expression of genes involved in inhibiting inflammatory response [78], an inhibition of expression of genes involved in inflammation (including several pro-inflammatory cytokines). GCs also have various non-genomic effects on cells via regulation of homeostasis of intracellular calcium [79], generation of reactive oxygen species, or regulation of the pathways involved in inflammation or apoptosis [77]. Regarding the effects of GCs on immune response, these can be separated into effects on innate immunity and effects on adaptive immunity, with repercussions for anti-tumour immune response (Figure 3).

### 2.1. Effect of Glucocorticoids on Innate Immunity

GCs act as powerful modulators of inflammation since they are involved in three essential stages of inflammation, namely alarm, mobilization, and resolution. During the alarm phase, tissue macrophages, mast cells, and stromal cells secrete inflammatory mediators, such as lipid agents, vasoactive amines, and cytokines, after signalling activation via pattern recognition receptors (PRR), which bind danger signals to their receptor. GCs inhibit the downstream signalling pathways of numerous danger signals, thereby supressing the production of inflammatory mediators. For example, GCs can inhibit the signalling of Toll-like receptors (TLRs) at several levels, namely by binding the GCs ligand receptors leading to inhibition of transcription factors, such as NF-κB, activator protein 1 (AP1), or IRF3. GCs also promote expression of inhibitors of TLR signalling, such as Dual specificity protein phosphatase 1 (*DUSP1*), which attenuates the activity of kinase protein 1 activated by MAP kinases (MAPK1), and of IL-1R associated kinase 3 (IRAK-3) [80]. In addition, GCs also induce the leucine zipper protein (*GILZ*; also known as *TSC22* domain family protein 3), which in turn strongly inhibits NF-κB [81]. The inhibition of certain TLR signalling factors has knock-on effects on other signalling pathways of inflammatory mediators, such as cytokines, which leads to suppression of cytokine production, especially pro-inflammatory cytokines, such as IL-1 beta, TNF alpha, interleukins (IL) 2, 3, 4, 5, 10, and 12, and interferon (IFN)γ [82]. During this phase, GCs can also inhibit the release of histamine by mast cells, thereby making it possible to limit the allergic reaction [83].

During the mobilisation phase, GCs modulate the production of lipid mediators, such as prostaglandins and leukotrienes, thereby preventing vessel dilation and permeability [84,85]. GCs further act on recruitment and adhesion of leukocytes by inhibiting adhesion molecules (integrins and selectins) [86,87].

During the resolution of inflammation, GCs stimulate the differentiation of macrophages towards the M2 phenotype, which in turn produce TGF and IL-10, both cytokines implicated in the resolution of inflammation, enabling a return to the baseline state [88,89].

### 2.2. Effect of Glucocorticoids on Adaptive Immunity

A key element in cellular immunity is the activation of T-lymphocytes specific to the antigen mediated by the T-cell receptor. Antigen-presenting cells (APC) present antigens as peptides bound to major histocompatibility complex (MHC) class I or class II molecules to active CD8+ or CD4+ T-lymphocytes, respectively. GCs play an important role in adaptive immunity, notably in proliferation, differentiation, and functionality of T cells. In the context of glioblastoma, a decrease in the number of intratumoral immune cells, specifically T cells, was demonstrated after dexamethasone treatment in a murine model [90].

T-cell activation is the result of a chain of events, namely progressive maturation of dendritic cells that can efficiently present the antigen, followed by interaction between the co-stimulatory CD28 molecule located on the CD4+ cell and CD80 located on the dendritic cell (second, co-stimulatory signal). Next, the dendritic cells present antigenic peptides in association with MHC class II to the T-cell receptor on the T lymphocyte. Finally, cytokines are activated, such IL-12 and TNF-α. GCs can modulate the activation of T cells by dendritic cells via their impact on dendritic cell activity, notably by inhibiting their maturation [91] but also by down-regulating expression of MHC class II molecules [92], CD1a lipid-presenting molecules, co-stimulatory molecules (e.g., CD80 and CD86), and pro-inflammatory cytokines (e.g., IL-12 and TNF) while simultaneously promoting expression of anti-inflammatory cytokines (e.g., IL-10) [93].

Furthermore, GCs can also inhibit the activation of T cells via the TCR signalling pathway, which is induced by the interaction between the MHC and the TCR. The mechanisms involved include down-regulation of c-Fos expression and inhibition of activator protein-1 (AP-1), NF-κB, and nuclear factor of activated T cells (NF-AT) [94]. This leads to reductions in the proliferation and secretory capacity of cytokines, such as IL-2 by T-lymphocytes [95].

In parallel, GCs affect the polarisation of naïve T cells by preventing the differentiation of naïve T cells into Th1 cells via inhibition of IL-12 production by APCs, thus promoting polarisation and response by Th2 and regulatory T cells (T-regs) [96]. This phenomenon is associated with a reduction in expression of the Th1-lineage-specific T-bet and strongly enhanced expression of Th2-associated cytokines, such as IL-4, IL-10, and IL-13 [97]. Regarding polarisation of T-regs to the detriment of Th1 response, it was shown in a mouse model that GCs treatment led to a significant increase in transient *FOXP3* mRNA expression by CD4 + T cells. Accordingly, in the setting of asthma, the proliferation and circulation of T-regs was augmented in relation to Th1 cells [98]. In addition, GCs inhibit Th17 polarisation via expression of the *GILZ* protein, which suppresses the factors capable of inducing Th17 lymphocytes (i.e., IL-1, IL-6, IL-23 dendritic cells), and expression of genes implicated in differentiation and activity of Th17 cells (IL-17A, IL-23 receptor, RORγt, STAT3, BATF, and IRF4) [99,100].

### 2.3. Effects of Glucocorticoids on Anti-Tumoral Immunity

Among the important biological characteristics of tumours is the capacity of tumour cells to escape the immune system [101]. Immunosurveillance seeks to repress the development and proliferation of cells that are recognized as abnormal by the immune system. It involves both innate and adaptive immunity. In view of the predominantly inhibitory action of GCs on immune response (particularly cell response and Th1), it appears of prime importance to evaluate the capacity of GCs to interfere with an efficacious anti-tumour immune response. Paradoxically, outside the context of treatment with immunotherapy (see below), the impact of GCs use on immune response in patients with cancer has not been widely studied. Much of the available biological data comes from preclinical studies. However, it should be underlined that, contrary to other immunosuppressive medical therapies, treatment with GCs has never yet been associated with a significant increase in the occurrence of cancer in humans [102,103]. Conversely, it has been clearly demonstrated that in healthy volunteers, the administration of dexamethasone led to a rapid reduction in the number of circulating T cells to reach a nadir 4 to 8 h after injection, with a rebound in T cells above baseline at 24 h after hydrocortisone infusion [104]. More pronounced lymphopenia has also been observed in patients receiving dexamethasone versus those not receiving it in several series of patients treated for cancer, particularly cerebral tumours [105,106]. Lymphodepletion appears to be more marked in the peripheral compartments compared to intratumoral compartments [91,107,108] and is mainly the result of apoptosis [90] as shown by the expression of late apoptosis markers as early as 1 h after GCs administration. The effect of GCs on lymphocyte apoptosis persisted in these murine models after repeated doses of dexamethasone [90]. Dexamethasone administration was also shown to reduce NK cells and myeloid populations (particularly activated cells), in support of a general inhibitory effect on anti-tumour immune response in tumour-bearing mice [90].

These quantitative effects are also accompanied by qualitative effects on CD4+ and CD8+ T cells, with inhibition of the proliferation and differentiation of naïve T cells but without an equivalent effect on memory T cells [109]. Again in murine models, treatment with dexamethasone was shown to reduce the capacity of CD4+ and CD8+ T cells to mount an effective Th1-mediated response [90].

GCs further appear to play a role in expression of ICIs. Accordingly, the study by Giles et al., showed that dexamethasone treatment led to upregulation of CTLA-4 expression in CD4 and CD8 T cells associated with alteration of CD28 co-stimulation in human T cells and a mouse model. CTLA-4 blockade was shown to partially rescue T cell number [109]. In mouse models, dexamethasone enhanced PD-1 expression during T-cell activation in a dose-dependent manner. This effect is mediated by GR signalling since the addition of the GR antagonist mifepristone blocked induction of the PD-1 receptor [110].

More recently, other preclinical studies have shown in mouse models that there is a gradient of increasing GR expression and signalling from naïve through to terminally dysfunctional CD8+ tumour-infiltrating lymphocytes (TILs) [111]. In this latter study, repeated exposure of CD8+ T cells to dexamethasone not only deeply inhibited pro-inflammatory cytokine production but also rapidly induced multiple checkpoint inhibitors on the cells (including PD-1, Tim-3, Lag-3). All these effects were related to activation of the GC receptor, and the presence of active glucocorticoid signalling was associated with poor response to immune checkpoint blockade [111]. These preclinical data are in line with observations in patients with melanoma, in whom a GCs activation signature was associated with failure of immunotherapy [112]. These biological effects observed in mouse models and partially confirmed in humans likely explain certain clinical findings, such as the fact that patients treated with dexamethasone for cerebral tumours were unable to generate an immune response after anti-tumour neo-antigen vaccination contrary to patients who received no dexamethasone [113].

The deleterious effects of GCs on immune response are clearly well established in preclinical models, and there exists a body of data to support an identical role in humans, especially in patients with cancer, but the magnitude of the effect remains unclear as of yet. The recent advent of new immunotherapies using ICIs has nurtured the debate about the potentially deleterious effects of GCs in this treatment setting.

## 3. Challenges of Corticosteroid Use in the Immunotherapy Era

Over the last few years, ICIs have revolutionized treatment and become the first-line metastatic treatment in many forms of cancer. New immunotherapies are mainly represented by monoclonal anti-PD1, anti-PD-L1, or anti-CTLA-4 antibodies. They can be used in monotherapy or in combination with each other or with other therapeutic classes, such as chemotherapy [114,115,116]; with other targeted therapies, such as tyrosine kinase inhibitors; or other monoclonal antibodies. 

Given the potentially deleterious effects of GCs on the quality of anti-tumour immune response, it is crucial to understand the pharmacological effects of GCs in patients concomitantly receiving immunotherapy. In particular, an improved understanding of the impact of GCs in the setting of immunotherapy would make it possible to better manage their use in certain situations that are frequent in cancer patients, such as to treat immune-mediated adverse events, to manage cancer-related symptoms, or for prophylaxis of adverse effects of chemotherapy associated with immunotherapy (Figure 4). 

### 3.1. GCs to Manage Immune-Related Adverse Events Caused by Immunotherapy

In parallel to the use of GCs to manage or prevent side effects of chemotherapy, the advent of immunotherapy has led to oncologists observing a rise in different side effects, which are immune-mediated. These so-called “immune-related adverse events” (irAEs) arise from the mechanism of action of ICIs, notably by reactivation of exhausted T cells, which can lead to a loss of self-tolerance and thus to the onset of certain auto-immune manifestations [117].

The range of irAEs is wide and varied, and there is a growing body of literature describing them thanks to the numerous randomized trials of ICI [118,119,120,121]. IrAEs occur more frequently with anti-CTLA-4 than with anti-PD-(L)1 in monotherapy [122]. They can affect numerous organs, including the thyroid [123], skin [124], colon, liver, or lungs [125]. The intensity of irAEs also varies: in the least severe cases, they may not require any therapeutic intervention, whereas the most severe cases can lead to hospitalization, intensive care with life-support therapies, or even death [126].

Management of irAEs comprises the temporary or definitive discontinuation of the immunotherapy and introduction of immunomodulatory therapy, namely GCs as first-line choice (recommended for the majority of irAEs of grade 2 or higher), thanks to their multiple anti-inflammatory and immunosuppressant properties as previously described [93,127,128]. The dose and duration of GCs treatment will depend mainly on the affected organ and the severity of symptoms [129]. The use of GCs in this context is now well standardized thanks to guidelines from professional societies in oncology [130,131,132]. In the majority of cases, the initial prescription of GCs should lead to the rapid resolution of the irAE, and then, GCs should be tapered off and ultimately discontinued. In case of irAEs that resist GCs treatment, more powerful immunomodulatory or immunosuppressant therapies may be required [129,133].

Nevertheless, although GC treatment could decrease or abrogate irAEs, it is possible that the immune response will not be entirely inhibited. GCs have major inhibitory effects on naive CD8+ T cells but little impact on the proliferation and activity of activated CD8 T cells. This phenomenon could be explained by the increased production of IL-2 by proliferating CD8 T cells, leading to GC resistance [134]. However, steroid-refractory irAEs are rather uncommon and require a more precise understanding of the physiopathology as underlined previously [135]. To avoid a prolonged duration of GC therapy (especially when given in high doses) and to avoid the risk of GC-resistant mechanisms in case of severe irAEs, some teams have suggested immunosuppressive therapy, such as TNF-alpha inhibitors, as first-line therapy [136]. Nevertheless, nowadays, GC therapy remains the first-line treatment for irAES in current guidelines.

### 3.2. Influence of Corticosteroids on the Efficacy of Immunotherapy

Due to their effects on innate and adaptive immunity [93], it is legitimate to question the possible consequences on immunotherapy efficacy of prescribing GCs during the management of irAEs.

Accordingly, some studies have reported that corticosteroid use could be associated with lower efficacy of immunotherapy. Firstly, in the setting of metastatic non-small-cell lung cancer (mNSCLC), Arbour et al., investigated the impact of GCs treatment at the time of initiation of immunotherapy with anti-PD-1 or anti-PD-L1 [137]. In independent cohorts, they observed that baseline corticosteroid treatment with ≥10mg of prednisone equivalent daily at the start of PD-(L)1 blockade was associated with reduced survival (overall response rate, progression-free survival, and overall survival). These findings prompted the authors to recommend prudent use of corticosteroids at the time of initiating PD-(L)1 blockade. Similarly, less favourable clinical outcomes and lower overall survival were reported in patients with NSCLC treated with nivolumab and concurrent GCs [138].

In the setting of cerebral tumours, dexamethasone is frequently used, as mentioned above, to treatment symptoms related to cerebral oedema. In this context, Iorgulescu et al., investigated the dose-dependent effect of GCs on the progression of glioblastoma in patients receiving immunotherapy [90]. Baseline dexamethasone in patients with PD-1 blockade was associated with survival. Interestingly, the deleterious effect seemed to be dose-dependent in this study. Indeed, dexamethasone was associated with alterations in both innate and adaptive immunity, with a reduction in the number of myeloid and NK cells, an increase in apoptosis, and a reduction in T-cell function. By multivariate analysis, baseline dexamethasone administration was the strongest predictor of poor survival. Similar results have also been described in metastatic melanoma, where a negative association between baseline corticosteroids and overall survival was reported [139].

However, available data are conflicting, and a recent review of the literature, including 27 studies, concluded that the concomitant administration of corticosteroids and ICIs may not necessarily lead to poorer clinical outcomes although it was not possible to defined a dose or exposure threshold beyond which corticosteroids might be associated with poorer immunotherapy efficacy [140]. In line with this, a pooled analysis of four studies, including two phase 3 trials of immunotherapy in patients with advanced melanoma, failed to find a deleterious effect of GCs on the efficacy of treatment [141]. In metastatic renal cell carcinoma, the largest cohort of patients treated in routine practice with nivolumab (an anti-PD-1) in the GETUG-AFU-26 NIVOREN trial [142] compared patients taking ≥ 10 mg of prednisone equivalent prior to progression under immunotherapy with those not exposed to steroids and failed to find a difference in progression-free or overall survival between the two groups [143].

These results are concordant with the conclusions of Horvat et al., who reviewed the effect of corticosteroids on overall survival and time to treatment failure in patients receiving ipilimumab, outside of clinical trials, for metastatic melanoma [144]. Given the frequency of irAEs under ipilimumab, treated patients often received GCs (35%) for the management of these events, but no difference in survival or time to treatment failure was observed between those who required systemic corticosteroids and those who did not. Additionally, in the setting of metastatic melanoma treated with ipilimumab, Downey et al., reported that the duration of tumour response was not affected by the use of high-dose steroids to abrogate treatment-related toxicities [145].

Taken together, these results nevertheless seems to suggest a difference between exposure to GCs therapy prior to immunotherapy initiation and exposure to GCs for the treatment of irAEs [146]. Exposure to GCs before introduction of ICIs or early after ICI initiation appears to be more often associated with poorer response.

Accordingly, in patients with mNSCLC, several studies have shown a deleterious effect of early steroid use (within 30 days prior to initiation of ICIs) as compared with steroids initiated at a later phase of ICI treatment [138,147]. Early corticosteroid therapy is frequently present in patients with poor general status, more advanced disease, and in those with cerebral metastasis [147]. In a meta-analysis including 16 studies [148], steroids initiated to mitigate irAEs did not affect overall survival, whereas patients taking steroids for cancer-related symptoms had poorer overall and progression-free survival. Similarly, other studies [138,147] have also suggested that the use of GCs (usually for cancer-related symptom control) prior to initiation of immunotherapy is associated with poorer disease control and shorter progression-free and overall survival. Early GCs use is often associated with lower performance status and more extensive disease. Therefore, confounding factors affecting survival exist in these patients that may partially explain the worse outcomes observed with immunotherapy when initiated in patients with previous or early GCs use. This was clearly highlighted in the study by Ricciuti et al., showing that in patients treated with ICIs for NSCLC, poor outcomes were driven mainly by a subgroup of patients receiving corticosteroids for palliative indications to control cancer-related symptoms [149]. In any case, these results, taken together, suggest that the lowest possible dose of GCs should be used at initiation of immunotherapy.

Regarding patients who receive GCs after immunotherapy has been initiated, the majority are patients in whom irAEs emerge. Several studies, notably in lung cancer or melanoma, suggest an association between the occurrence of irAEs and improved survival under immunotherapy [119,150,151]. Indeed, irAEs could be the reflection of a stronger immune response, which is simultaneously associated with greater anti-tumour activity. Accordingly, in patients with renal cell carcinoma, Verzoni et al., observed improved overall survival at one year in patients with irAEs compared to those without adverse immune-induced reactions [152]. Similar findings were observed in patients with mNSCLC [118]. However, these results should be interpreted with caution, as they may simply be the result of immortality bias, whereby patients who live longer are treated longer and thus have a higher probability of eventually presenting irAEs. With regard to the doses of GCs used to mitigate irAEs, literature data are conflicting. Several studies [144,153] suggest that the temporary use of high GCs doses for the management of irAEs may not negatively influence survival.

Conversely, Faje et al., reported that the use of low-dose GCs to treat ipilimumab-induced hypophysitis was associated with longer survival, compared to high-dose GCs [154].

### 3.3. Role of Corticosteroids in Immunotherapy-Chemotherapy Associations

In numerous cancers, immunotherapy is now widely used on top of other therapies, notably cytotoxic chemotherapy, in the hope of achieving therapeutic synergy in patients who are sensitive to both treatments. There exists a biological rationale to support the idea that some chemotherapies may have favourable immunological effects [155,156]. The positive effects of chemotherapeutic drugs on the immune system are mainly mediated by selective depletion of regulatory and immunosuppressive cells, such as T-regs (with cyclophosphamide) [157] or myeloid-derived suppressor cells (gemcitabine, 5-fluorouracil) [158,159]. Another potential mechanisms is the induction of immunogenic cell death in tumour cells, thereby stimulating an immune response against the tumour (with anthracyclines or oxaliplatin) [160,161,162].

In this context, it remains unclear what the impact on the immune system may be of repeated administration of GCs, notably to mitigate allergies or vomiting, on top of an association combining chemotherapy and immunotherapy. Combinations of chemotherapy plus immunotherapy have become the standard first-line treatment in mNSCLC [116,163], oesophageal-gastric cancer [164], as well as in triple-negative breast cancer both in the neoadjuvant [165] and metastatic settings [166]. The cytotoxic drugs commonly used in these associations are mainly platinum salts, which are highly emetogenic, or taxanes (paclitaxel, docetaxel), which frequently cause allergic reactions and thus theoretically justify pre-treatment regularly provided in the form of GCs.

Regarding the potential impact of GCs on the efficacy of associations combining a taxane and immunotherapy, discordant results have been reported. For example, in metastatic triple-negative breast cancer, the IMpassion 130 study [167] associating the anti-PD-L1 atezolizumab with chemotherapy by nab-paclitaxel (a taxane not requiring anti-allergy premedication by GCs) showed a benefit in terms of progression-free and overall survival. Conversely, the IMpassion 131 study [168] conducted in the same patient population also using atezolizumab but together with paclitaxel (which does require anti-allergy pre-treatment by GCs) found no survival benefit of adding the ICI on top of paclitaxel. The reasons for these discrepancies remain to be elucidated, but the hypothesis of a negative impact of GCs in patients receiving paclitaxel has been advanced. Therefore, these findings should prompt systematic consideration about the appropriate and reasonable use of GCs when immunotherapy is associated with chemotherapy. In this regard, the results of a study by Lansinger et al., are of prime importance: in a chart review of over 3000 patients who received steroid premedication (predominantly dexamethasone) prior to chemotherapy by paclitaxel or docetaxel [169], the authors found no correlation between dexamethasone dose or route and subsequent hypersensitivity reactions. These findings could be determinant for the issuance of guidelines for the use of GCs in the context of chemo-immunotherapy combinations. Furthermore, it should be noted that the conflicting results with atezolizumab between nab-paclitaxel and paclitaxel were not replicated in other studies [170], including in metastatic triple-negative breast cancer [166].

It thus seems clear that pending further data, caution should be advised, and the lowest possible dose of steroids should be used. Lansinger et al., suggested that a dose of 10 mg of dexamethasone is sufficient to limit hypersensitivity reactions occurring with taxanes [169].

## 4. Conclusions

GCs are one of the most widely used treatments in patients with cancer for the management of numerous cancer-related symptoms or to prevent or mitigate adverse side effects occurring during cancer therapy. The impact of GCs on anti-tumour immune response was long neglected and not taken into consideration. The almost daily use of GCs in oncology is a position that may now need to be revised in light of our improved understanding of their deleterious pleiotropic effects on anti-tumour immune response, particularly in patients treated by ICIs.

The real impact of GCs use on the efficacy of immunotherapy (alone or in association) remains poorly understood but seems to depend not only on the dose but also on the therapeutic indication for GCs and the timing of their introduction with regard to initiation of immunotherapy.

Pending further clinical results, it is mandatory to systematically weigh the potential risks and benefits for every indication for corticosteroid therapy, especially in patients receiving immunotherapy, with a view to prescribing the lowest GCs doses and for the shortest.

## Figures and Tables

**Figure 1 cells-11-00770-f001:**
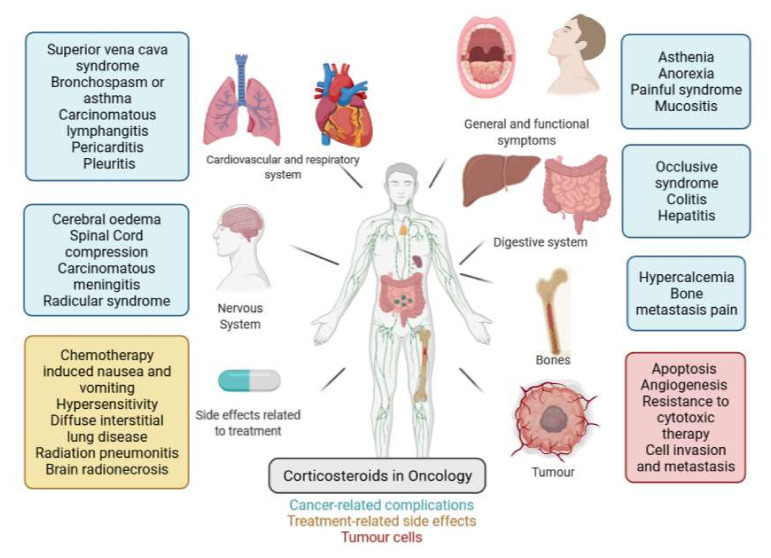
Indications for corticosteroid therapy in oncology (created with BioRender).

**Figure 2 cells-11-00770-f002:**
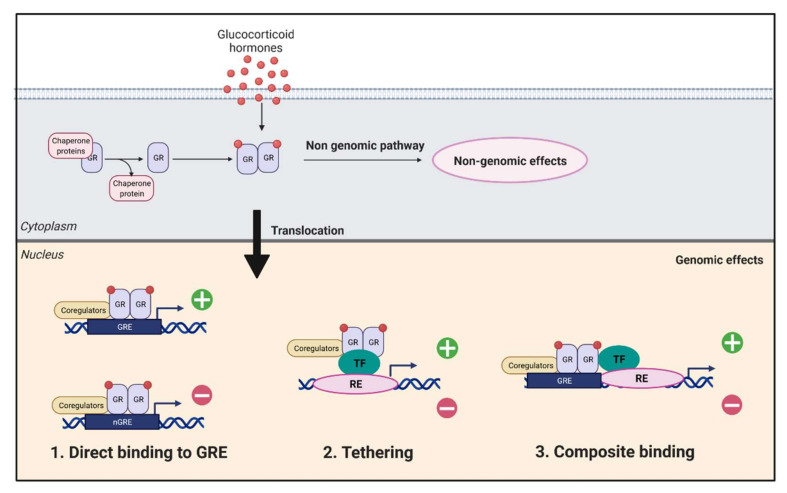
Mechanisms of glucocorticoids activity. Glucocorticoids will penetrate the plasma membrane to the cytoplasm, where they will have genomic and non-genomic effects. Following the binding of the hormone on its receptor (GR), the complex will be translocated into the nucleus and will have multiple mechanisms. (**1**) The binding of the complex on the response element of glucocorticoids (GRE), which will allow the expression or repression of target genes. (**2**) The complex binds to a transcription factor (TF) located on its response element (RE) in order to prevent or activate transcription. (**3**) Binding of the complex to DNA and protein substrates to prevent or activate transcription (created with BioRender).

**Figure 3 cells-11-00770-f003:**
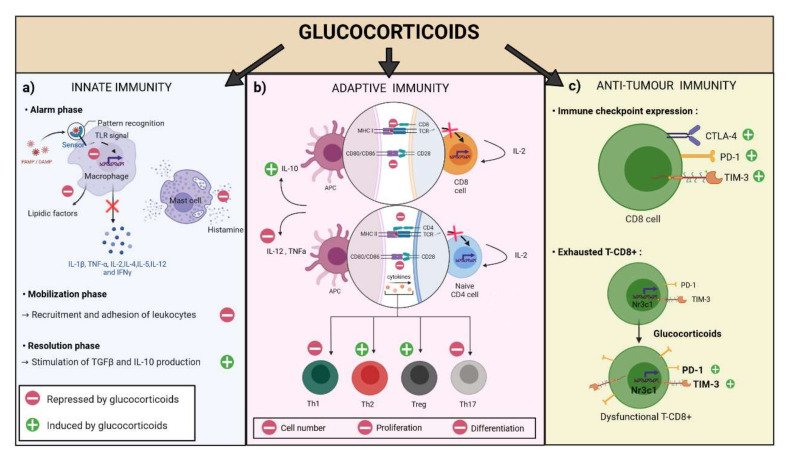
Effect of glucocorticoids on the immune system. Glucocorticoids have a role on innate, adaptive, and antitumour immunity. (**a**) On innate immunity, during the alarm phase, they inhibit toll like receptor (TLR) signalling, which prevents the production of proinflammatory cytokines. In addition, they inhibit the release of histamine from mast cells. The mobilization phase is impacted by a decrease in leukocytes recruitment and adhesion. The resolution phase is characterized by an increased production of TGFβ (Transforming growth factor β) and interleukin 10 (IL-10). (**b**) On adaptive immunity, glucocorticoids play an important role because they inhibit the co-stimulation of the antigen-presenting cell (APC) to T-CD8+ or T-CD4+ lymphocytes by decreasing the expression of major histocompatibility complex (MHC) type I and II molecules and decreasing the expression of CD28 and CD80 molecules. In addition, T-cell proliferation is decreased by alteration of T-cell receptor (TCR)-initiated signalling and thus decreased IL-2 production. Glucocorticoids also have a role on the polarization and differentiation of naive CD4-T cells by inhibiting T helper 1 (Th1) and 17 (Th17) differentiation and promoting T helper 2 (Th2) and T regulator (T-reg) differentiation. (**c**) On anti-tumour immunity, glucocorticoids induce the expression of immune checkpoints, such as the cytotoxic T-cell associated protein 4 (CTLA4), programmed death receptor 1 (PD-1), and mucin-3 T-cell immunoglobulin (Tim-3). They also promote an exhausted and therefore dysfunctional phenotype of CD8+ lymphocytes through the co-expression of PD-1 and TIM-3 receptors (created with BioRender).

**Figure 4 cells-11-00770-f004:**
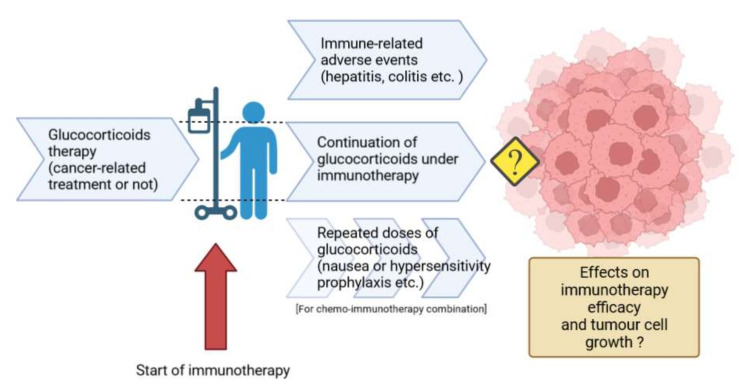
Challenge of glucocorticoid therapy in patients receiving immunotherapy (created with BioRender).

**Table 1 cells-11-00770-t001:** Summarize of effects of glucocorticoids on tumour cells.

Pathways	Targets	Effects	Tumour Types
Apoptosis	Bcl2 miR-17~9 ROS (reactive oxygen species)	Increased	Haemopathic malignancies
	BIM, BAX, BAK TXNIP, GILZ miR-708	Decreased	Haemopathic malignancies and solid tumours
Proliferation	AP1, Nf-κB c-MYC	Decreased	Haemopathic malignancies and solid tumours
Invasion/migration	SGK1 Selectins E-cadherin	Decreased	Solid tumours
	RhoA MMP2/9 IL-6	Decreased	Solid tumours
Angiogenesis	VEGF/IL-8	Decreased	Solid tumours
Resistance to cytotoxic therapy	SGK1 MKP1 (DUSP1) IκBα	Increased	Solid tumours

## Data Availability

Not concerned. Data sharing not applicable.

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
