# Peer review of "Impact of Glucocorticoid Use in Oncology in the Immunotherapy Era"

_cells, 2022, doi:10.3390/cells11050770_

Round 1
Reviewer 1 Report
There are some typos in figure 1 in"General and functionnal symptoms" and "nuceleus" in line 181
Reviewer 2 Report
The authors submitted fine review considering use of GCs in cancer treatment.
Minor concerns.
- I think that adding figure or table summarizing effect of glucocorticoids on tumor cells will make this paper more reader friendly
- Please clarify which of the pro or anti-tumour effects of GCs are specific for leukemia/lymphoma and for solid tumors
- Please add more data about glucocorticoid-induced apoptosis and resistance in neoplastic cells
- Please check again use of abbreviations GCs and GC. In my opinion it needs some corrections.
- Please correct IL6, IL8 etc to IL-6, IL-8…
Reviewer 3 Report
In this manuscript, the authors described the therapeutic effects of glucocorticoids (GCs) on cancer from both clinical and molecular biological aspects. This article is well written and summarized, providing interesting and important information on the clinical use of GCs for cancer treatment. My comments and questions are as follows:
- Abstract and Introduction: The authors should mention irAE.
- Background: In this section, the authors put too much emphasis on clinical aspects of GCs. The readers of "Cells" are more likely interested in the molecular mechanisms of GCs, like those described in lines 55–81. Although these clinical outcomes are interesting, the authors should also explain molecular mechanisms how GCs elicit these effects.
- In line with my comment above, target molecules of GCs in the setting of cancer therapy should be summarized.
- lines 81–87: Are there any arguments in the beneficial effects of GCs on hematological malignancies?
- lines 187–190: Are these effects (interaction with other transcription factors) ligand-dependent?
- lines 388–390: If it has been reported regarding the GCs-resistant mechanisms of irAEs, please briefly explain it.
- line 180: Nr3c1 –> NR3C1 (italicized). Please see: https://www.genenames.org/about/guidelines/
Round 2
Reviewer 3 Report
The authors have adequately responded to my comments.
Author Response
Thank you for your feedback.